# Simulation models of sugary drink policies: A scoping review

**Natalie Riva Smith**[1], **Anna H. Grummon**[2,3], **Shu Wen Ng**[4,5], **Sarah Towner Wright**[6], **Leah Frerichs**[7]*

1 Department of Social and Behavioral Sciences, Harvard TH Chan School of Public Health, Boston, MA, United States of America, 2 Department of Nutrition, Harvard TH Chan School of Public Health, Boston, MA, United States of America, 3 Department of Population Medicine, Harvard Medical School / Harvard Pilgrim Health Care Institute, Boston, MA, United States of America, 4 Department of Nutrition, Gillings School of Global Public Health, Chapel Hill, NC, United States of America, 5 Carolina Population Center, UNC Chapel Hill, Chapel Hill, NC, United States of America, 6 Health Sciences Library, UNC Chapel Hill, Chapel Hill, NC, United States of America, 7 Department of Health Policy and Management, Gillings School of Global Public Health, Chapel Hill, NC, United States of America

* leahf@email.unc.edu

**Data Availability Statement:** All relevant data are within the manuscript and its Supporting Information files.

**Funding:** NRS was supported by T32-HD091058, P2C-HD050924, and T32-CA057711 of the

## Abstract

### Introduction

Simulation modeling methods are an increasingly common tool for projecting the potential health effects of policies to decrease sugar-sweetened beverage (SSB) intake. However, it remains unknown which SSB policies are understudied and how simulation modeling methods could be improved. To inform next steps, we conducted a scoping review to characterize the (1) policies considered and (2) major characteristics of SSB simulation models.

### Methods

We systematically searched 7 electronic databases in 2020, updated in 2021. Two investigators independently screened articles to identify peer-reviewed research using simulation modeling to project the impact of SSB policies on health outcomes. One investigator extracted information about policies considered and key characteristics of models from the full text of included articles. Data were analyzed in 2021–22.

### Results

Sixty-one articles were included. Of these, 50 simulated at least one tax policy, most often an *ad valorem* tax (e.g., 20% tax, n = 25) or volumetric tax (e.g., 1 cent-per-fluid-ounce tax, n = 23). Non-tax policies examined included bans on SSB purchases (n = 5), mandatory reformulation (n = 3), warning labels (n = 2), and portion size policies (n = 2). Policies were typically modeled in populations accounting for age and gender or sex attributes. Most studies focused on weight-related outcomes (n = 54), used cohort, lifetable, or microsimulation modeling methods (n = 34), conducted sensitivity or uncertainty analyses (n = 56), and included supplementary materials (n = 54). Few studies included stakeholders at any point in their process (n = 9) or provided replication code/data (n = 8).

National Institutes of Health (https://www.nih.gov/). LF was supported by K01-HL138159 (https://www.nih.gov/). AHG was supported by T32-HL098048 and K01-HL158608 (https://www.nih.gov/). The funders had no role in study design, data collection and analysis, decision to publish, or preparation of the manuscript. National Institutes of Health, T32-HD091058, Natalie R Smith; National Institutes of Health, P2C-HD050924, Natalie R Smith; National Institutes of Health, T32-CA057711, Natalie R Smith; National Institutes of Health, K01-HL138159, Leah Frerichs; National Institutes of Health, T32-HL098048, Anna H Grummon; National Institutes of Health, K01-HL158608, Anna H Grummon.

**Competing interests:** The authors have no competing interests to declare.

## Discussion

Most simulation modeling of SSB policies has focused on tax policies and has been limited in its exploration of heterogenous impacts across population groups. Future research would benefit from refined policy and implementation scenario specifications, thorough assessments of the equity impacts of policies using established methods, and standardized reporting to improve transparency and consistency.

## Introduction

Overconsumption of sugar-sweetened beverages (SSBs) is a key contributor to high and rising cases of non-communicable diseases worldwide [1, 2]. Experts agree that policy action is needed to reduce SSB consumption and prevent diet-related disease [3]. For example, the World Health Organization has called for countries to tax SSBs as one way to reduce SSB consumption [1]. Other policy options include front-of-package warning labels, limits to portion sizes, and marketing restrictions [3, 4].

Decision makers often want to consider and compare the consequences of proposed policy designs before implementation. Simulation modeling is a powerful tool for projecting likely population health outcomes under different policy scenarios. Broadly, models use existing knowledge and data to project how consumer and supply-side behaviors (e.g., SSB consumption, product reformulation) and health outcomes (e.g., obesity, diabetes) are likely to change over time in response to policy actions [5]. Modifying model parameters allows investigators to examine different 'what if' scenarios, such as how expected health impacts might differ if the policy was less effective, or if consumers or suppliers respond in particular ways. This functionality makes simulation modeling a compelling method for providing policymakers with information on the likely health outcomes of different policy actions to inform policy design and implementation.

A growing number of studies have used simulation models to project how SSB policy action might impact population health outcomes. To advance SSB policy research with simulation modeling, it is important to synthesize trends in the type and amount of evidence available across these studies and identify areas for improvement. Prior reviews have examined simulation models of nutrition policies generally [6–9], but have not focused specifically on SSB policies, despite their growing importance. Other work has reviewed the effects of specific SSB policies like taxes [10] or warning labels [11], without a focus on simulation modeling studies exclusively. What is missing from the current literature is a clear understanding of the variety of SSB-specific policies that have been assessed with simulation models, and the characteristics of the models.

Thus, we aimed to conduct a systematic scoping literature review to describe the current state of the SSB policy simulation modeling literature. The goal of this review was to spur thoughtful considerations of next steps for simulation modeling of SSB policies, including where policy evidence might be lacking and where methodologies can be improved. We focused on two questions: 1) what SSB policies have been evaluated using simulation modeling and 2) what are the characteristics of the simulation models used, including the models' settings/populations, health outcomes, and modeling methods?

## Methods

We used systematic scoping review methods, as our research questions were related to the broad scope of literature on SSB policy simulation models [12, 13]. Scoping reviews are broader in scope than traditional systematic reviews, but like systematic reviews, scoping reviews define eligibility criteria, systematically search the literature, and extract data from included studies [14]. A trained clinical health sciences librarian (STW) performed a systematic electronic search of publications in PubMed, Cumulative Index to Nursing and Allied Health Literature (CINAHL) via EBSCO, EMBASE via Elsevier, PsycInfo via EBSCO, Cochrane Central Register of Controlled Trials, SCOPUS, and Communication and Mass Media Complete via EBSCO, collecting results from the inception of the database through June 25, 2020. A database search update was performed on June 10, 2021. Our search terms addressed the three main concepts of the review: 1) computer simulation or computer model or economic evaluation; 2) sugar-sweetened beverages; and 3) health policy or public health or nutrition guidelines (S1 File). We included articles that used mathematical simulation modeling in a human population, presented novel findings, simulated at least one policy focused exclusively on SSBs, translated policy impacts to health outcomes beyond behavior change, and were published in English. We excluded economic modeling that simulated changes in consumption only, without translating consumption changes into health outcomes (e.g., demand system modeling [15]). We also excluded articles that included an SSB policy as one component of a multi-faceted intervention or policy (e.g., a three-component childcare intervention to increase physical activity, reduce screen time, and replace SSBs with water [16]), unless SSB-exclusive policies were examined in comparison to these multi-component policies. We also excluded articles targeting sugar consumption generally, not specifically sugar consumption from SSBs (e.g., added sugar labeling policies [17]). We used Covidence software (Veritas Health Innovation, Melbourne, Victoria, Australia) to screen abstracts and full-text articles [18].

Two investigators (NRS and LF) independently screened abstracts; AHG resolved discrepancies at this stage. The same two investigators then independently screened full text articles for inclusion and resolved discrepancies through discussion. In addition to articles identified by the database search, we included an article known to our team that was not picked up by search terms because it did not have an abstract [19]. We also screened the reference lists of full text articles from 2020 and 2021. NRS extracted data from included articles in Redcap [20] using a standardized extraction template and a random 10% of article extractions were checked by the senior author. We extracted data using only the main text of articles for all questions except for one question specifically about the inclusion of a supplement and its level of detail. We did not infer details beyond what was explicitly stated by authors.

## Results

### Included articles

The database search yielded 4,903 titles/abstracts after excluding duplicates (Fig 1) [21]. Of these, 4,761 were excluded during abstract screening, leaving 142 full text articles assessed for eligibility. Sixty of these were eligible for inclusion [22–81]. In the full text review stage, articles were most often excluded for not examining at least one policy focused exclusively on SSBs (n = 32) or because they were a conference abstract (n = 22). We include 61 articles in our results, adding in the article known to our team that did not include an abstract to the 60 identified via database and reference list searches [19].

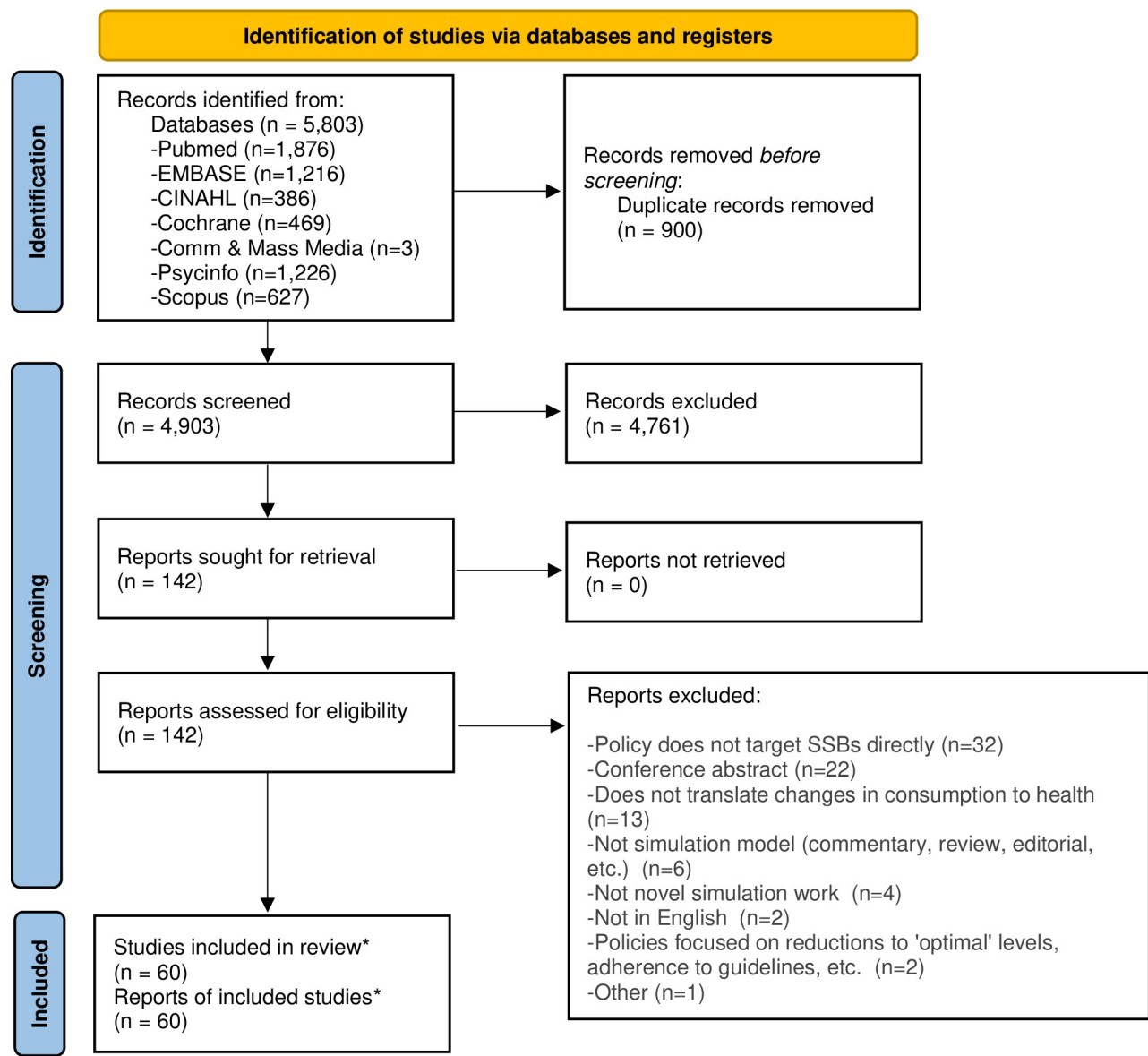

**Fig 1. PRISMA 2020 flow diagram for new systematic reviews.** Notes: *An additional article known to our team that was not picked up by search terms because it did not have an abstract was added after identification of articles via registers and databases, bringing the final number of studies included to 61.

Included articles were published between 2011 and 2021, with over half published in 2017–2021 (n = 37). The main text provides aggregate statistics on the included articles; individual data for each can be found in S1 Table and in an interactive table at https://natsmith. shinyapps.io/Article-Information/.

## RQ1: What policies have been evaluated using simulation modeling?

Most articles (n = 54) simulated one SSB policy. Six articles simulated two SSB-focused policies, and one simulated three. Some articles simulated SSB policies in comparison to other health promotion policies (n = 10). For example, Basu *et al.* 2013 simulated a ban on SSB purchasing with nutrition assistance dollars and a one-cent-per-fluid-ounce SSB volumetric excise

tax (two SSB-focused policies) alongside two fruit and vegetable incentive policies (other health promotion policies) [27].

Fifty out of 61 included articles examined at least one tax policy (Fig 2). To fully characterize taxes, we extracted information on both the tax rate (e.g., *ad valorem*/percentage-based, such as 20%, or unit-based, such as 1 cent-per-fluid-ounce) and how that tax rate would be implemented (i.e., excise, sales, other, unclear), based on the exact language used in the article [3, 5]. *Ad valorem* tax rates were the most commonly examined tax (n = 25), with most studies of tax policies examining a 20% tax on SSBs (n = 20/25). Volumetric taxes were the second most common (n = 23, also referred to as volume-based taxes); 13 of these evaluated 1-cent-per-fluid-ounce taxes. Among articles that described tax implementation, most evaluated taxes were implemented as excise taxes; however, many articles did not specify implementation mechanisms (Fig 2). Two articles noted that they were specifically not discussing tax implementation mechanisms to minimize the modeling assumptions required. Studies generally simulated impacts on consumption by first estimating changes in SSB prices under tax policies, using assumptions about baseline prices of SSBs and the percent of tax passed through (i.e., the amount of tax that the taxed entity 'passes through' to consumers via price increases). Changes in prices were then translated to changes in consumption using price elasticities (which quantify the percent change in consumption for a percentage change in price).

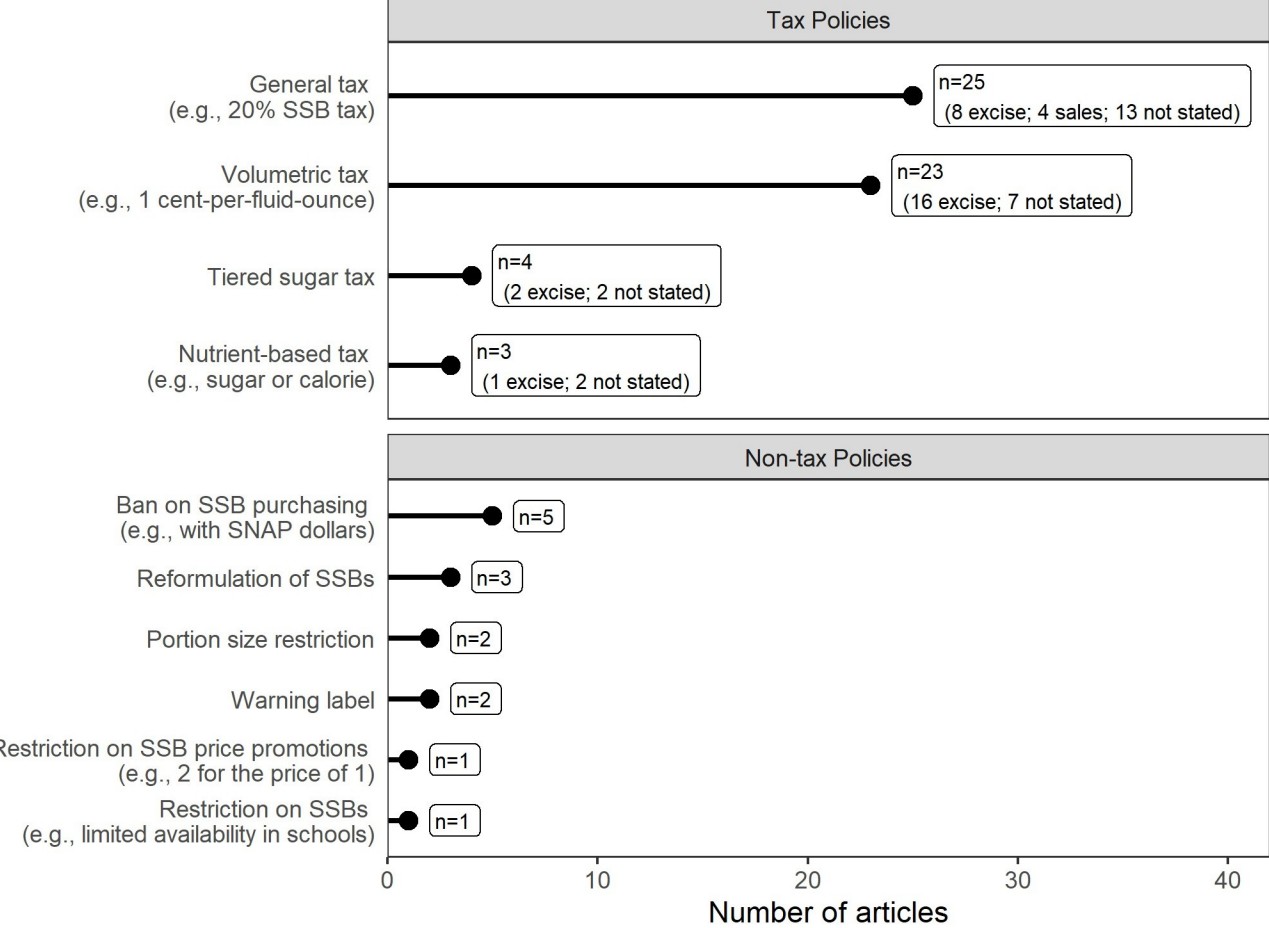

**Fig 2. Sugar-sweetened beverage policies examined by simulation modeling studies (n = 61).** Notes: SNAP = Supplemental Nutrition Assistance Program.

Thirteen of the 61 articles simulated a non-tax policy (two simulated both a tax and non-tax policy, Fig 2). Five studies modeled bans on SSB purchases. Policies to ban SSBs most commonly focused on prohibiting SSB purchases using US Supplemental Nutrition Assistance Program benefits. Articles also examined policies designed to restrict the use of price promotions for SSBs (n = 1, stores could not sell SSBs under 'two-for-one' deals) or restrict the availability of SSBs in schools (n = 1). Two studies, both in the US, examined policies requiring warning labels on SSBs. Two studies examined portion size policies that would prohibit the sale of SSBs larger than 375 milliliters (about 12.7 fluid ounces) or 250 milliliters (about 8.5 fluid ounces). Finally, three studies considered policies requiring reformulation targets for SSBs (e.g., policies requiring manufacturers to reduce added sugars in SSBs by a given percentage). Simulations of these policies used estimated impacts from published behavioral science research or assumptions about behavioral responses to these changes. For example, a warning label simulation model used observed effects on purchases of SSBs from a randomized trial in a mock convenience store [43, 82]. Another model simulated the consumption effects of a portion size policy by assuming any modeled individual who drank a beverage larger than the portion size cap in the policy would reduce their consumption after policy implementation to drink exactly the portion size specified in the policy [35].

## RQ2: What are the characteristics of SSB policy simulation models?

A wide range of countries were represented in the included texts, with the US being the most commonly modeled country (n = 24, Table 1). To simulate potential health effects of policies in these countries, models use hypothetical populations with characteristics that are similar to the population of interest (e.g., adults in the US, children aged 5–18 in Australia). Nearly all studies modeled hypothetical populations with age (n = 59) and gender or sex attributes (n = 58, Table 1). Studies generally presented results by population subgroups (n = 48), which can shed light on a policy's potential to affect disparities in health outcomes (specific subgroups examined by included studies are shown in Table 1).

Table 1 also displays major health outcomes simulated. All but 7 studies translated changes in SSB consumption into impacts on weight (n = 54), typically using energy balance approaches [83–89]. These approaches translate changes in energy intake (e.g., decreases in calories from SSBs under a policy change) into changes in body weight [83]. Forty studies used some form of an energy balance equation (or heuristic based on an energy balance equation) in their modeling approach, with equations from Hall *et al.* being the most commonly used (n = 24/40). Notably, five studies assumed that eating 3,500 fewer calories equates to 1 pound of weight lost, an energy balance heuristic that has been widely criticized [83, 90–92]. Nine studies used estimates of direct effects of SSB intake on weight change from published literature.

Language used to describe modeling methods varied widely (Table 2). When stated, the most commonly used simulation methods were cohort models (Markov or life table modeling, n = 6 and 15, respectively) or microsimulation models (n = 13). Six studies stated that they used comparative risk assessment methods, two used system dynamics modeling, and two used agent-based modeling.

Studies typically simulated outcomes over a 10-year (n = 18), 20-year (n = 5), or lifetime (n = 14) time horizon. In some cases (n = 14) the time horizon was not clearly stated. Nearly half of the studies stated that their work was based on an existing model or modeling framework (n = 26, e.g., ACES-Obesity [93], CHOICES [94], CVD-PREDICT [95]). Thirty of the included studies included a visual of their logic model or modeling flow. Forty-six articles included a descriptive table of input parameters, though the specific format of tables varied

**Table 1. Populations and outcomes modeled in included studies (n = 61).**

| Variable | | N | % |
|---|---|---:|---|
| Countries modeled | | | |
| | US | 24 | 39% |
| | Australia | 8 | 13% |
| | Mexico | 5 | 8% |
| | South Africa | 4 | 7% |
| | UK | 3 | 5% |
| | All other countries[a] | 17 | 28% |
| Attributes given to simulated populations[b] | | | |
| | Age | 59 | 97% |
| | Sex or gender | 58 | 95% |
| | Income | 21 | 34% |
| | Race, ethnicity, nativity, or related | 14 | 23% |
| | Education | 4 | 7% |
| | SNAP | 4 | 7% |
| | Socioeconomic status | 4 | 7% |
| Attributes for results stratification (n = 48 out of 61 that stratified results)[b] | | | |
| | Age | 33 | 69% |
| | Sex or gender | 26 | 54% |
| | Income | 18 | 38% |
| | Race, ethnicity, nativity, or related | 12 | 25% |
| | Socioeconomic status | 2 | 4% |
| Outcome[b] | | | |
| | Weight or BMI | 54 | 89% |
| | Diabetes | 30 | 49% |
| | Cardiovascular disease | 24 | 39% |
| | Cancer | 12 | 20% |
| | Dental caries | 7 | 11% |
| | Osteoarthritis | 8 | 13% |
| | Kidney disease | 2 | 3% |
| | Quality of life outcome[c] | 20 | 33% |
| | Economic outcome[d] | 36 | 59% |

Notes: US = United States, UK = United Kingdom, SNAP = Supplemental Nutrition Assistance Program, BMI = Body Mass Index.

[a]Other countries simulated in fewer than 3 studies include Germany (n = 2), Thailand (n = 2), Canada (n = 1), Colombia (n = 1), Ecuador (n = 1), England (n = 1), Global (n = 1), India (n = 1), Indonesia (n = 1), Ireland (n = 1), Netherlands (n = 1), New Zealand (n = 1), Philippines (n = 1), Portugal (n = 1), Zambia (n = 1).

[b]Articles could simulate more than one attribute or outcome, so percentages will not sum to 100.

[c]For example, quality-adjusted life years.

[d]For example, disease-attributable healthcare costs, cost-effectiveness ratios.

widely. Some articles presented high-level overviews and included information like data citations or distributional assumptions (e.g., Lal *et al.*, 2017 [49]). Other articles presented more granular information on specific parameters such as average SSB consumption among different age groups (e.g., Ma *et al.*, 2016 [56]). All studies mentioned assumptions of their work, and most studies performed some form of sensitivity or uncertainty analysis (n = 56).

**Table 2. Modeling methods of included studies (n = 61).**

| Variable | | N | % |
|---|---|---:|---|
| Modeling Methods | | | |
| | Life table modeling | 15 | 25% |
| | Microsimulation | 13 | 21% |
| | Markov cohort modeling | 6 | 10% |
| | Comparative risk assessment | 6 | 10% |
| | System dynamics modeling | 2 | 3% |
| | Agent-based modeling | 2 | 3% |
| | Other or not stated | 17 | 28% |
| Time Horizon[a] | | | |
| | 10 years | 18 | 29% |
| | 20 years | 5 | 8% |
| | Lifetime | 14 | 23% |
| | Unclear | 14 | 23% |
| | Other (e.g., 1 year, 50 years) | 13 | 21% |
| Methods Details | | | |
| | Existing model or modeling framework | 26 | 43% |
| | Visual of modeling flow or logic | 30 | 49% |
| | Table of input parameters | 46 | 75% |
| | Assumptions mentioned | 61 | 100% |
| | Included sensitivity or uncertainty analyses | 56 | 92% |
| | Supplementary materials | 54 | 89% |
| | Replication code, pseudocode, or data provided | 8 | 13% |
| | Included stakeholders | 9 | 15% |

Notes: [a]Articles could simulate over multiple primary time horizons (e.g., 10 years and over the cohort lifetime), so percentages will not sum to 100.

Most studies included supplemental files (n = 54) with varying levels of detail. Twenty supplemental files only presented additional tables/figures, without any further exposition on the modeling methods. Particularly useful appendices included detailed descriptions of how the authors came to modeling decisions (e.g., Wilde *et al.*, 2019 [80]) or how a method was implemented (e.g., Basu *et al.*, 2013 [27]). Although supplements were quite common, including data or code to replicate models was much less common (n = 8). Examples of methods for providing replication code included posting datasets in publicly accessible repositories and discussing equations and pseudocode (i.e., narrative/plain language description of computer code) in supplemental files [22], or providing code directly on GitHub [25].

Stakeholder involvement–of any stakeholder, at any time in modeling work–was described by 9 studies. For example, Urwannachotima *et al.* engaged stakeholders in exercises to help build the structure of their model [77]. Models from the CHOICES group in the US incorporate stakeholder input into their intervention selection and implementation considerations [42, 54, 55].

## Discussion

We identified 61 studies that used simulation modeling methods to project the potential health impacts of policies targeting SSB consumption. Use of simulation models to evaluate SSB policies has grown over time; all studies were published after 2011, and over half were published

within the past four years (2017–2021). Consistent with prior literature, we find that the most commonly evaluated SSB policy is a tax, with the tax literature dominated by *ad valorem* and volumetric tax policies [8, 9]. We also document an emerging literature that includes other policy options such as purchasing bans, warning labels, and portion size restrictions. Most models we reviewed used either cohort or microsimulation modeling methods, simulated a population defined by age and gender or sex, and projected changes in weight, diabetes, or cardiovascular disease. These results are in line with findings from other reviews of food policies [8, 9]. Our results point to norms in the literature and highlight areas for future work to build on this strong foundation.

A closer examination of the articles revealed that future policy design and dissemination work would benefit from models including more explicit details about policy design and implementation. For example, some articles examining taxes modeled only on the final price change in SSBs induced by the tax. This could be problematic because some tax designs can have markedly different impacts on SSB consumption and health outcomes [19, 51, 96], even when they raise prices by the same amount [19]. For example, one study found that taxing sugar content instead of beverage volume would increase the public health benefits of an SSB tax by 30% because sugar-based taxes could create price incentives for consumers to substitute from higher- to lower-sugar SSBs, while volumetric taxes would not [19]. Additionally, many articles we reviewed did not specify how a given tax rate would be implemented. This could lead to inaccurate or imprecise results from simulation models because, for example, research shows that consumers tend to respond less strongly to taxes that are added at the register (e.g., sales taxes) compared to those reflected in the shelf price (e.g., excise taxes) [97]. In the case of excise taxes, strategic responses by manufacturers or distributors may also result in differential price-pass through of the tax and/or reformulations to minimize the tax (under sugar-based taxes) across their product portfolios and their market shares or dominance in product categories which could vary geographically [96, 98, 99]. New evidence also suggests that the way shelf prices show (or do not show) the inclusion of an SSB tax also impacts efficacy [100].

Researchers could also provide more policy details and implementation scenarios around non-tax policies, which would provide valuable implementation advice for policymakers. For example, when evaluating a warning label policy, the topic (health or nutrient warning) and design (text or graphic) of the warning label used to develop estimates of efficacy should be specified. These details are important because nutrient warnings have been shown to generate substantial product reformulation as companies seek to reduce nutrient densities to below the regulatory thresholds that trigger warnings [101, 102]; these supply-side changes are likely to amplify demand-side effects of warnings and should be incorporated into simulation models of warning policies. For policies such as portion size restrictions, clearly defining what SSBs would be targeted and where restrictions would be in place is critical; observational and experimental research also indicates that focusing on unsealed drinks sold in food service establishments, targeting large drinks sold at convenience stores, or limiting free refills can greatly impact potential reductions in consumption and health outcomes [103–105].

Broadly, future modeling research should seek to be attentive to real-world policymaking and implementation questions. Modeling results will be more useful for policy implementation when researchers clearly define tax and non-tax policies and include implementation details in their models, including the scope of regulated SSBs and associated implementation scenarios. Models can be used to probe how different contexts impact policy implementation, or how industry responses to policy implementation could impact policies' realized health effects [33]. With an eye towards informing policymaking and implementation, engaging stakeholders will be critical to ensure that models have the best chance to inform advocacy efforts and contribute to policymaking and implementation.

We found that all articles discussed the assumptions their model made, and nearly all reported some form of sensitivity or uncertainty analyses, though the descriptions of such analyses and language used varied widely. Future work should build from this base to include more concrete discussions of how assumptions, and their potential violations, might impact results, and should be specific about the strength of parameter estimate(s) used. Including these details is important both to establish confidence in modeled results (e.g., if there are concerns about the causal strength or appropriateness of parameter estimates used) [106] and to help policymakers understand what to expect under different implementation scenarios [107]. For example, included studies evaluating excise tax policies often assumed a 100% pass-through rate in their primary models and examined results assuming alternative rates in sensitivity analyses. This approach is useful and could be strengthened by linking results to a discussion of when and why we might expect pass through rates to vary (for example, based on different implementation considerations or industry responses given known market concentration). The SSB modeling literature would also benefit from using methods such as probabilistic Value of Information (VOI) analyses [108, 109] which offer a structured way to prioritize research dollars towards future behavioral science or policy research that would reduce decision uncertainty.

Most models we reviewed focused on one policy. An important next step will be for researchers to simulate multiple policy options within one modeling framework to compare policy effectiveness, and possibly expand into comparing policy options with other types of public health action such as community-based programs or interventions. Comparative assessments can help policymakers considering multiple policy options identify tradeoffs given potential limited resources for implementation and limited political capital, potentially making research more useful to policymakers and increasing its use in policy decision making. Modeling also offers a way to anticipate the potential impacts of multi-policy, multi-sectoral obesity and chronic disease prevention plans [110]. Modeling multiple policies could also help researchers uncover potential interactions between policies, though additional behavioral science research will be needed to support this by providing evidence on how consumers respond to different combinations of policies (e.g., warning labels combined with taxes) [111].

Most studies we reviewed modeled SSB policy impacts on weight, diabetes, cardiovascular diseases (including stroke and hypertension), and cancers. Emerging work has considered additional health outcomes, including dental caries, kidney disease, and osteoarthritis. Models typically presented population average outcomes alongside outcomes by subgroups, with most focused on age or sex or gender groups and fewer studies evaluating results by income, race, ethnicity, education, or other sociodemographic characteristics. Future research should continue to include individual heterogeneity to paint a more complete picture of policies' potential to affect health equity. Researchers should also consider methods specifically designed to consider equitable impacts of policies, particularly those drawn from the field of economic evaluation [112–115] such as equity-based weighting, extended cost-effectiveness analysis, distributional cost-effectiveness analysis, and multi-criteria decision analysis [112]. For example, equity-based weighting involves increasing (or decreasing) the contribution of outcomes for different groups (e.g., increasing the weight of quality-adjusted life years gained among low-income cohorts or individuals) [112, 114]. As authors seek to further consider equity implications, techniques like microsimulation models that allow for using distributions from the relevant population(s) in question will become increasingly important [9].

Future research should consider a number of other improvements to modeling methods. For example, methods like agent-based and system dynamics modeling allow analysts to incorporate important complexity when studying SSB policies, such as interactions between individuals and their social and physical environments and feedback loops between health

behaviors [116]. Applying these methods to SSB policies is a fruitful new area of research, as the models we reviewed generally did not consider how policy impacts may differ due to social network effects. Failing to account for how social relationships may relate to food consumption [117], other health behaviors [118, 119], and downstream health effects like weight [120–122] could lead to estimates of policy impact, both overall and within subgroups, that are over- or under-estimated.

Another area for improvement is replicability. Very few articles we reviewed provided code or data to replicate their work, and supplementary material often focused more on supplementary results rather than providing additional methodological details that would support replication. We advocate for increased transparency and code sharing of simulation models, as other reviews have called for [8, 9]. For example, researchers should consider the framework set out by Alarid-Escudero and colleagues [123] and utilize platforms such as GitHub or the Open Science Framework.

Standardized reporting guidelines for simulation modeling could also help push the field towards more consistent and transparent modeling [8, 9]. In our study, data extraction was at times challenging due the many disciplines (e.g., health economics, epidemiology, behavioral science) represented in this research. Although multi-disciplinarity offers many benefits, the diversity of disciplines engaging in SSB policy modeling also led to articles using different simulation vocabulary, informal reporting norms (e.g., what details are reported in the main text versus supplementary material), and formal reporting requirements (e.g., journal word and figure limits). Past research has provided guidance for improving modeling research practices [124], but to our knowledge there are no standardized systems for reporting on simulation models. Existing guidelines are either focused on specific types of modeling [125, 126] or economic evaluation more broadly [127, 128]. The CHEERS checklist, for example, is targeted towards economic evaluations but lacks specific guidance for simulation models [127, 128]. Reporting guidelines for simulation modeling could set out common language for discussing sensitivity and uncertainty analyses, specify what methods details should be in the main text of an article versus in supplementary material (e.g., time horizon, time step used for discrete models), and set standards for reporting and discussing model assumptions. Given the large number of analytic decisions involved in developing a simulation model, clear guidance about what to report is critical for building confidence in published models, creating comparability across models, and helping researchers make better *a priori* decisions. While the specific details relevant to different kinds of models may differ (e.g., there is no specific time component in comparative risk assessment models [129]) reporting guidelines will help make these differences clear.

## Limitations

As with any review, we may have missed relevant articles in our search. However, we built a comprehensive and systematic search along with a trained information specialist, and used terms similar to previously published reviews of simulation modeling [130] and SSB warning labels [11]. Our inclusion/exclusion criteria enabled us to include a range of studies, yielding a comprehensive commentary on the state of the science and allowing us to identify important considerations for future SSB policy simulation modeling. Although errors may have been made in the data extraction process, we used a standardized extraction template to ensure consistency between articles and a random 10% of article extractions were checked by the senior author.

## Conclusions

Simulation modeling is a powerful tool for projecting how SSB policies could impact public health. Many SSB policies have shown potential for improving population-level health, but

decision making requires more specific and nuanced understanding of policy effects. Our review indicates key areas for improvements in simulation modeling methods, including that future work should incorporate more details regarding how policies would be implemented, thoroughly assess the equity impacts of policies using established methods, and standardize reporting to improve transparency and consistency. These improvements will lead to higher-quality simulation models that better inform public health decision making.

## Supporting information

**S1 File. Database specific search terms.**
(PDF)

**S2 File. PRISMA-ScR checklist.**
(PDF)

**S3 File. Full text exclusions.**
(XLSX)

**S1 Table. Individual article information.**
(DOCX)

## Author Contributions

**Conceptualization:** Natalie Riva Smith, Anna H. Grummon, Shu Wen Ng, Leah Frerichs.

**Data curation:** Natalie Riva Smith, Sarah Towner Wright, Leah Frerichs.

**Formal analysis:** Natalie Riva Smith.

**Investigation:** Natalie Riva Smith.

**Methodology:** Natalie Riva Smith, Anna H. Grummon, Sarah Towner Wright, Leah Frerichs.

**Project administration:** Natalie Riva Smith.

**Supervision:** Shu Wen Ng, Leah Frerichs.

**Visualization:** Natalie Riva Smith.

**Writing – original draft:** Natalie Riva Smith.

**Writing – review & editing:** Natalie Riva Smith, Anna H. Grummon, Shu Wen Ng, Sarah Towner Wright, Leah Frerichs.

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
