## [Decision Letter · Decision Letter 0]

19 Jul 2022

PONE-D-22-09129Simulation models of sugary drink policies: A scoping reviewPLOS ONE

Dear Dr. Frerichs,

Thank you for submitting your manuscript to PLOS ONE. After careful consideration, we feel that it has merit but does not fully meet PLOS ONE’s publication criteria as it currently stands. Therefore, we invite you to submit a revised version of the manuscript that addresses the points raised during the review process.

We look forward to receiving your revised manuscript.

Kind regards,

Louisa Ells, Ph.D.

Academic Editor

PLOS ONE

Journal Requirements:

Reviewers' comments:

Reviewer's Responses to Questions

**Comments to the Author**

1. Is the manuscript technically sound, and do the data support the conclusions?

Reviewer #1: Yes

Reviewer #2: Yes

2. Has the statistical analysis been performed appropriately and rigorously? 

Reviewer #1: Yes

Reviewer #2: Yes

3. Have the authors made all data underlying the findings in their manuscript fully available?

Reviewer #1: Yes

Reviewer #2: Yes

4. Is the manuscript presented in an intelligible fashion and written in standard English?

Reviewer #1: Yes

Reviewer #2: Yes

5. Review Comments to the Author

Reviewer #1: In the search terms, 'modeling' may also be spelled as 'modelling' with two 'l's. You may want to make greater use of the asterisk '*' for expanding search terms in future.

On page 6, "we included an article known to our team that was not picked up by search terms because it did not have an abstract". It would be interesting to know why this wasn't found by keyword matches in the title, and if you considered amending your search strategy. Perhaps there are other articles without abstracts that will have been excluded?

Also this article is referenced on page 7 (80) but perhaps reference it on page 6 as well, where you mention it for the first time.

Tangent: I have been working on my own scoping review recently and I constructed a 'test set' of papers and tested my search strategy against this set to see if it retrieved all the papers I wanted. This might be an interesting technique for you in future reviews.

Thank you for publishing your code and your extracted data! :) This is very useful for other people who want to dig into your results more.

I would also suggest publishing the original results of your database searches, and including a column to indicate at which stage each result was excluded. This would allow readers to gain more insight into your screening process, perform additional checks, and mine your results for more specific or slightly different literature / research questions. I don't know if you can export this data from Covidence but it might be a quick win if possible. I would also include data on exclusions at the full text/eligibility stage as well.

The interactive summary table on shinyapps.io is a really nice touch! :)

Page 8 uses the phrase "ad valorem tax" which is not a term I have come across. Perhaps a short parenthetical to explain what these are like "(value added taxes which are set at a fixed percentage of the final sale price)" would be good.

In table 1, it is not clear to me what "quality outcome" means. Please can you add a description or clarify? Does this mean "(health related) quality of life"?

Thank you for reporing on the publication of code and data in these models, it is something I care deeply about! What a shame only 8 papers published their code!

The stakeholder involvement comment is also very interesting!

On page 13 you mention the increase in publication over time. Perhaps a line chart of cumulative published studies over time" would be interesting?

On page 13 you state "different tax rates – such as volume-based vs. sugar-based taxes – can have markedly different impacts on SSB consumption and health outcomes, even if they are economically equivalent" which points to other drivers of change apart from price signalling. This sounds very counterintuitive, and it's not immediately clear to me what these other drivers would be (perhaps reformulations or repackaging?). Please could you add an example to this paragraph to show how two different types of tax, at the same equivalent rate, could lead to different consumption / health outcomes?

Throughout the article you have sprinkled in critical reflections on the literature you have reviewed and this is really fantastic, it makes the paper much more informative and interesting to read than just a 'simple' review. :)

On page 14 you discuss "pass-through rates" but again I'm not sure what this means. Please could you add a short parenthetical?

On page 15 you discuss some of the benefits of modelling multiple policy options in a single framework. You may also want to mention the needs of policy-makers to evaluate multiple potential options and any negative ans spillover effects, and that by presenting evidence on multiple options, researchers can make their outputs more useful to policy-makers and improve the uptake of their evidence into policy.

Thank you also for you emphasis on health equity and the need to examine different (intersectional) population cohorts. Again this is something policy-makers want to know, and it's also such an important part of any health policy research!

Overall an enjoyable and informative paper which has provided a service to the research and policy communities by drawing together the extant literature! :)

Reviewer #2: I found this article pleasant to read and informative. I have no major comments or concerns, although I do welcome greater clarity and emphasis through summary – perhaps a table – surrounding each study’s causal focus, quantification of potential biases, and mitigation of these potential biases within implementation for each simulation study.

This is covered comprehensively within the text but adding some explicit causal inference terminology and providing an overall summary might be quite a powerful message. It is because I worry that most scientific studies (whether engaging in simulation or not) seek causal understanding yet do not always explicitly state this; and, whether they do or not, most studies undertake insufficient (or no) evaluation of the robustness of their findings (beyond the usual caveats stated about study limitations in the discussion).

I do not suggest this to be onerous, as the information is mostly there already, but I would welcome clarity on these issues through a summary of all studies with respect to: which were explicit in their pursuit of causal inference, which stated their causal estimand(s), and the extent to which each sought to assess or quantify robustness of study estimates (e.g., through sensitivity analyses or through any other form of quantitative bias analysis).

Along with these methodological perspectives, it would then be useful to summarize how uncertainties were related to stakeholder challenges of implementation (i.e., the degree by which implementation of policy in different contexts might need to mitigate against potential uncertainties in impact). This is already well described (e.g., lack of details on how tax rates would be implemented, affecting robustness of the simulation models), but being non-specific with respect to each policy consideration, a summary of all studies on the extent by which issues of implementation in relation to model uncertainties are considered and addressed would be useful.

Such an overarching summary would then reflect the extent by which simulation studies do or do not fully embrace complexities in policy implementation across different settings, and reveal to what degree simulation studies are mindful of uncertainties in relation to implementation by first recognizing such an issue and then dealing with it as part of the modelling process, with potential advice for stakeholders engaged in subsequent implementation.

6. PLOS authors have the option to publish the peer review history of their article (what does this mean?). If published, this will include your full peer review and any attached files.

Reviewer #1: **Yes: **John Liam Preston

Reviewer #2: No

---

## [Author Response · Author response to Decision Letter 0]

29 Aug 2022

Please find attached response memo for our specific response to reviewer and editor comments.

---

## [Decision Letter · Decision Letter 1]

13 Sep 2022

Simulation models of sugary drink policies: A scoping review

PONE-D-22-09129R1

Dear Dr. Frerichs,

Thank you for submitting the revised version of your manuscript. All concerns of the reviewers have been appropriately addressed in the new revised manuscript and we thank you for your thorough responses to reviewers. We’re pleased to inform you that your manuscript has been judged scientifically suitable for publication and will be formally accepted for publication once it meets all outstanding technical requirements.

Kind regards,

Hans-Peter Kubis, PD. Dr. rer. nat.

Academic Editor

PLOS ONE

Additional Editor Comments (optional):

Reviewers' comments:

Reviewer's Responses to Questions

**Comments to the Author**

1. If the authors have adequately addressed your comments raised in a previous round of review and you feel that this manuscript is now acceptable for publication, you may indicate that here to bypass the “Comments to the Author” section, enter your conflict of interest statement in the “Confidential to Editor” section, and submit your "Accept" recommendation.

Reviewer #2: All comments have been addressed

Reviewer #3: (No Response)

2. Is the manuscript technically sound, and do the data support the conclusions?

Reviewer #2: Yes

Reviewer #3: Yes

3. Has the statistical analysis been performed appropriately and rigorously? 

Reviewer #2: Yes

Reviewer #3: Yes

4. Have the authors made all data underlying the findings in their manuscript fully available?

Reviewer #2: Yes

Reviewer #3: Yes

5. Is the manuscript presented in an intelligible fashion and written in standard English?

Reviewer #2: Yes

Reviewer #3: Yes

6. Review Comments to the Author

Reviewer #2: I am very satisfied with all the changes made and I have no further comments. The paper reads well and I recommend publication.

Reviewer #3: Simulation models of sugary drink policies: A scoping review

1. This article provides a scoping review of simulation models for policies to reduce SSB consumption. This is a revised submission after a previous round of comment. The revision appears responsive. The article is well-written, with insight into sound research methods. The topic is important.

2. The scoping review appears to address what policies are modeled and what methods are used, but not to cover what results these simulation studies found or how they affect real-world policy. That may be a reasonable limitation as a writing and publication matter, but it means the overall contribution to the literature is good rather than exceptional.

3. At first, in describing methods, I thought the article gave too little coverage to how uncertainty is addressed. But later the discussion section covers this issue adequately. Optionally, the authors may consider if this would have been good to incorporate more formally into the results section. In general, I think this entire field of simulation analysis depends heavily on assumptions that are themselves not necessarily well supported, so the importance of reporting uncertainty plainly could be emphasized even more strongly.

4. When the article discusses how such studies could better and more systematically make use of prior recommendations for best practice, another possible source is Neumann et al., Second Panel on Cost Effectiveness Methods. That tome is titled “cost effectiveness,” but as a stepping stone, some of its interior chapters may be a particularly good authoritative source on methods for these simulation studies.

5. The authors like advanced methods such as agent based modeling, but they also like replicable methods, which may sometimes imply simpler methods. In my view, the highly complex methods end up overstating the scientific basis of the findings and understating the sampling and non-sampling (modeling) sources of uncertainty, so I agreed more with the authors comments on replicability.

6. In the public health literature on this topic, researchers sometimes understate the importance of the supply function and over-emphasize the demand function. For example, they may recognize the importance of the own-price elasticity of demand but not talk explicitly about the supply elasticity. The fundamental economic issue is that the pass-through rate depends directly on the elasticities of supply and demand together. In this article, the authors come close to acknowledging the issue when they praise studies that simulate alternative assumptions about the pass-through rate, but it could have said even more clearly that this requires understanding the economics of SSB production and marketing not just consumer demand for SSBs.

7. PLOS authors have the option to publish the peer review history of their article (what does this mean?). If published, this will include your full peer review and any attached files.

Reviewer #2: No

Reviewer #3: **Yes: **Parke Wilde

---

## [Editor Report · Acceptance letter]

23 Sep 2022

PONE-D-22-09129R1 

Simulation models of sugary drink policies: A scoping review 

Dear Dr. Frerichs:

I'm pleased to inform you that your manuscript has been deemed suitable for publication in PLOS ONE. Congratulations! Your manuscript is now with our production department. 

Kind regards, 

on behalf of

Dr. Hans-Peter Kubis 

Academic Editor

PLOS ONE